# Comparison of Conservative Treatment of Cervical Intraepithelial Lesions with Imiquimod with Standard Excisional Technique Using LLETZ: A Randomized Controlled Trial

**DOI:** 10.3390/jcm10245777

**Published:** 2021-12-10

**Authors:** Andrej Cokan, Maja Pakiž, Tamara Serdinšek, Andraž Dovnik, Tatjana Kodrič, Alenka Repše Fokter, Rajko Kavalar, Igor But

**Affiliations:** 1Department for Gynaecological and Breast Oncology, University Medical Centre Maribor, Ljubljanska Ulica 5, 2000 Maribor, Slovenia; maja.pakiz@ukc-mb.si (M.P.); andrazdovnik@gmail.com (A.D.); tatjana.kodric@triera.net (T.K.); 2Department for General Gynaecology and Urogynaecology, University Medical Centre Maribor, Ljubljanska Ulica 5, 2000 Maribor, Slovenia; todorovic.tamara@gmail.com (T.S.); igor.but@ukc-mb.si (I.B.); 3Medical Faculty Maribor, Taborska Ulica 8, 2000 Maribor, Slovenia; alenka.repse-fokter@guest.arnes.si; 4General Hospital Celje, Oblakova Ulica 5, 3000 Celje, Slovenia; 5Department for Pathology, University Medical Centre Maribor, Ljubljanska Ulica 5, 2000 Maribor, Slovenia; rajko.kavalar@gmail.com

**Keywords:** imiquimod, LLETZ, randomized controlled trial, squamous intraepithelial lesion

## Abstract

(1) Background: There are limited data on the success of conservative treatment of high-grade cervical squamous intraepithelial lesions (HSIL) with imiquimod directly compared to standard of treatment with LLETZ. (2) Methods: Patients aged 18–40 with histological HSIL (with high-grade cervical intraepithelial neoplasia, CIN2p16+ and CIN3), were randomly assigned to treatment with imiquimod or LLETZ. The primary outcome was defined as the absence of HSIL after either treatment modality. The secondary outcomes were the occurrence of side effects. (3) Results: 52 patients were allocated in each group and were similar regarding baseline characteristics. In the imiquimod group, 82.7% of patients completed treatment, which was successful in 51.9%. All patients in the LLETZ group completed treatment, which was successful in 92.3% (*p* < 0.001). In the subgroup of CIN2p16+ patients, treatment with imiquimod was not inferior to LLETZ (73.9% vs. 84.2%, *p* = 0.477). During and after treatment, no cases of progression to cancer were observed. Side effects and severe side effects (local and systemic) were more prevalent in the imiquimod than in the LLETZ group (88.5% vs. 44.2% (*p*-value < 0.001) and 51.9% vs. 13.5% (*p*-value < 0.001), respectively). (4) Conclusion: Generally, in patients with HSIL, LLETZ remains the gold standard of treatment. However, in a subgroup analysis of patients with CIN2p16+, the success rate was comparable between the two treatment modalities. Due to the prevalence of side effects, the treatment compliance with imiquimod use may, however, present a clinically important issue.

## 1. Introduction

After the introduction of cervical cancer screening programs with earlier detection and treatment of precancerous lesions of squamous or glandular origin, the incidence of and mortality from cervical cancer dropped substantially [1,2,3]. In Slovenia, cervical cancer is no longer one of the ten most common cancers in women. The lowest age-standardized incidence rate of cervical cancer was recorded in 2014 (6.8/100,000) and the lowest mortality rate in 2016 (1.7/100,000) [4]. Women aged 20–64 are included in the screening program [5]. Based on pathological Papanicolaou (PAP) smear and/or positive human papillomavirus (HPV) testing, a colposcopy with biopsy is indicated. According to the new Lower Anogenital Squamous Terminology Standardization Project for HPV-Associated Lesions (LAST) classification, squamous intraepithelial lesions (SIL) are divided into low-grade lesions—LSIL (CIN1 and CIN2 p16 negative lesions) and high-grade lesions—HSIL (CIN2 p16 positive and CIN3 lesions) [6,7]. HSIL is invariably associated with an HPV infection, and its prevalence is the highest during the reproductive years [8]. It is usually treated with excision or ablation, although ablation is rarely used since it does not offer many additional advantages compared to excisional techniques [9]. The most commonly used excisional technique, large loop excision of the transformation zone (LLETZ), is, according to some studies, associated with premature labour, before 37 weeks (RR 1.75), and extremely premature labour, before 28 weeks of gestation (RR 2.23), and could also be associated with a higher subfertility rate and a higher rate of spontaneous abortion, although data are not always consistent [10,11,12,13,14,15]. Since premature delivery is one of the most important causes of perinatal morbidity and mortality, alternative conservative methods for SIL treatment are constantly being evaluated. The immunomodulator imiquimod is one of the main target compounds for treating SIL [16,17,18,19,20]. It is a toll-like receptor 7 (TLR7) agonist that acts locally and induces a cellular response which can aid in the regression or even eradication of HPV-associated lesions. Imiquimod is currently used for treating genital warts. In vitro studies have shown promising effects in the treatment of several diseases, such as melanoma, prostate cancer, endometrial cancer, endometriosis, and vulvar and cervical intraepithelial lesions as well as cervical cancer [21,22,23,24,25]. However, clinical studies are lacking. With this in mind, the aim of our study was to evaluate whether topical treatment of HSIL with imiquimod is comparable to standard treatment with LLETZ.

## 2. Materials and Methods

We designed a single-centre, prospective, randomized controlled trial, which was conducted at our department. The principles embodied in the Declaration of Helsinki were followed [26]. We developed the research protocol following the CONSORT guidelines [27]. The trial was registered in 2017, prior to patients’ enrolment, as a research project at our University Medical Centre (IRP 2017-01-02, 22/05/2017) and was also part of research programme P3-0327, funded by the National Research Agency (ARRS). The National Medical Ethics Committee of the Ministry of Health (0120-13/2017, KME 58/02/17, 10/05/2017) approved the trial before initiation. The trial was retrospectively registered on clinicaltrials.gov, registration number NCT04859361, 26/04/2021.

The primary objective of the study was to establish the efficacy of treatment with imiquimod (experimental arm) and compare it to that of the standard treatment with LLETZ (control arm). Treatment success as a primary outcome was defined as the absence of HSIL in both arms. Successful treatment in the experimental arm was defined as absence of histological HSIL in diagnostic biopsies at 20-week follow-up (4 weeks after treatment completion). In the control arm, successful treatment was defined as absence of cytological HSIL in cytology 6 months after LLETZ. The different outcome measures were based on moderate accuracy of PAP smear [28]. Therefore, in order to minimize potential progression of cervical disease to cancer, avoid LLETZ and possible overtreatment, as well as to assess the potential residual disease as accurately as possible, we performed a colposcopy with biopsies 4 weeks after treatment with imiquimod was completed. Follow-up after LLETZ was performed with cytology without HPV testing at 6 months after the treatment, which was in concordance with our national guidelines—biopsies were performed in case of clinically visible lesions [3]. Secondary outcomes of the study were the incidence and severity of the side effects in both groups, which were evaluated during and after treatment using the 5th version of the Common Terminology Criteria for Adverse Events (CTCAE) guidelines. Other secondary outcomes, namely (i) the need for treatment with LLETZ within two years after primary treatment with imiquimod in the experimental arm; (ii) retreatment with LLETZ two years after primary treatment with LLETZ in the control arm; (iii) the modulatory effect of imiquimod on immunoregulatory molecules, and (iv) HPV clearance after treatment with imiquimod are expected to be available in three years.

Enrolment of patients started in November 2018 and closed in April 2020. Because a negative HPV test is very rare in patients with HSIL and since, according to our guidelines, we do not test patients for high-risk HPV in cases of HSIL, the test was not performed prior to inclusion in the study [29]. Inclusion criteria were as follows: (i) newly diagnosed and previously untreated HSIL in women aged from 18 to 35 years or up to 40 years in case of nulliparity; (ii) adequate colposcopy (i.e., fully visible lesion and transformation zone); (iii) negative pregnancy test; (iv) safe contraception; and (v) freely given and signed informed consent. Exclusion criteria were as follows: (i) previously diagnosed HSIL or AIS; (ii) previous LLETZ or classical conization; (iii) concomitant vulvar or vaginal lesion or neoplasia; (iv) other malignancies; (v) insufficient colposcopy; (vi) pregnancy or lactation; (vii) known hypersensitivity to imiquimod; (viii) any known contraindications to immunotherapy; (ix) known HIV or acute or chronic hepatitis; (x) immune deficiency, or (xi) participation in any other ongoing clinical trial. Before inclusion in the study, all eligible patients were informed that LLETZ is the current standard treatment for HSIL and that treatment with imiquimod is still experimental. They were asked to provide written, informed consent. Figure 1 shows a flow chart of the patient enrolment and the follow-up process.

All consecutive patients who met the inclusion criteria and were willing to participate in the study were randomly assigned to one of the two parallel groups receiving either treatment with self-applied, topical imiquimod (5% cream, 250 mg of cream with 6.25 mg of active substance—one sachet of 5% cream) or standard treatment with LLETZ. For randomization, the envelope technique was used. At the first visit, we collected data regarding baseline characteristics (age, smoking, histopathological diagnosis); furthermore, a control colposcopy was performed by an experienced gynaecologist, working in the field of colposcopy and gynaecological oncology. In the experimental arm, the participants were instructed regarding the storage of the cream (room temperature, no direct sunlight) and the correct method of self-application. They were informed of the possible side effects, such as local (vaginal or vulvar inflammation and lower urinary tract symptoms) and systemic side effects (flu-like symptoms like fatigue, malaise, and fever), and were asked to monitor them if they appeared. Imiquimod was self-applied three times a week for 16 weeks (maximum allowed duration of therapy) before going to sleep, using an IVITA^®^ menstrual cup, which was filled with 250 mg of cream and inserted in the vagina for a duration of 6–8 h. Patients were advised not to have sexual intercourse on the nights when the menstrual cup was inserted, and they were instructed to remove the menstrual cup the following morning and shower afterwards. During the first three days of menstrual bleeding, they were instructed not to apply imiquimod or insert the menstrual cup. In case of severe side effects, the number of applications was reduced to twice per week (biweekly application), and if the side effects were still present, it was further reduced to once per week (weekly application). For maximum control of cervical disease, a control colposcopy with a PAP smear and a punch biopsy were scheduled at 10 weeks to rule out progression, and at 20 weeks after the start of the treatment, the procedures were repeated to evaluate its success. At 20 weeks, biopsies were performed at the locations where lesions were previously present, and if there were any new lesions present, additional biopsies were performed. In case of disease progression or persistence, treatment with LLETZ was offered. In the control arm, LLETZ was scheduled after the patient’s next menstrual period. LLETZ was performed in an outpatient setting with local anaesthesia, using KLS Martin Maxium with loop devices ranging from 10 to 20 mm in size. The excision was performed using monopolar current with cut frequency set to 100–150 W. Treatment success was evaluated in accordance with our national guidelines, i.e., 24 weeks after the procedure using a PAP smear with or without a punch biopsy [3]. All cytological and pathological specimens were prepared and analysed by a team of experienced cytologists and pathologists from the same hospital. To detect p16INK4a in formalin fixed paraffin embedded tissue (FFPET), sections of 4–5 μm were cut from uterine cervical mucosa samples, mounted on the adhesive slides, and subjected to immunohistochemistry (IHC) using CINtec p16 Histology Ventana. The IHC used an UltraView Universal DAB Detection Kit. Immunoperoxidase staining was performed on a BenchMark Ultra from Ventana Medical System (Roche Diagnostics, Basel, Switzerland). On-slide appropriate negative and positive controls were included. P16 positivity was defined as strong block-type staining involving basal keratinocytes and extending beyond the lower third of the epithelium.

### Study Power and Sample Size Calculation

Studies available at the time of protocol development demonstrated that imiquimod induced regression of cervical disease in 73%, and LLETZ in 95% of patients [16]. Using these regression rates, the desired 80% power of the study and an alpha of 5%, we calculated that, based on the primary objective and outcome measure, the sample size required 52 women in each arm. Calculations were performed using the G*Power application [30]. Statistical analysis was performed using SPSS Statistics software 25.0 (IBM, Armonk, NY, USA). Descriptive statistics were calculated on basic patient characteristics. Pearson’s chi-square/Fisher’s exact tests were used for comparison of categorical data between groups, and Student’s *t*-test for independent samples was used to compare normally distributed data between groups. Data was analysed using the intention-to-treat (ITT) principle, meaning that all enrolled patients were included in the final analysis, including those who did not completely adhere to the assigned treatment or did not complete it. The only exception from this principle was the evaluation of side effects in the experimental arm at 10 and 20 weeks after treatment initiation, where a per-protocol analysis was performed. A post hoc subgroup analysis was performed for patients with CIN2p16+ and CIN3 lesions. Statistical significance was set at a *p*-value < 0.05.

## 3. Results

During the enrolment period, 139 consecutive women were assessed for eligibility. Of these, invasion was suspected at the first visit in two patients; four did not meet the inclusion criteria, and 29 declined to participate. In total, 104 women (74.8% response rate) were included in the study, 52 in the experimental and 52 in the control arm (Figure 1). The treatment groups (imiquimod vs. LLETZ) were balanced with respect to baseline characteristics, such as age (28 ± 4.2 vs. 27 ± 4.6; *p*-value = 0.52), smoking (42.3% vs. 32.7%; *p*-value = 0.42), and histopathological diagnosis, which was divided into CIN2p16+ and CIN3 (44.2% vs. 36.5% and 55.8% vs. 63.5%; *p* = 0.55) (Table 1). No patients in either group received HPV vaccine.

In the imiquimod group, treatment was completed by 82.7% (43/52) of patients and was successful in 51.9% (27/52) of all patients or in 62.8% (27/43) of patients who completed the assigned treatment protocol (Table 2). In the CIN2p16+ and CIN3 subgroups, treatment was successful in 73.9% (17/23) and 34.5% (10/29) of patients, respectively. Side effects were present in 88.5% (46/52) of patients (Table 3, Appendix A). Moderate (grade 2) and severe (grade 3) side effects were present in 38.5% (20/52) and 13.5% (7/52) of patients, respectively (Table 3). When comparing side effects at scheduled follow-ups (at the 10th and the 20th week), there was a slight decrease in their overall occurrence (80.8% vs. 76.7%) as well as occurrence of moderate (34.6% vs. 32.6%) side effects. Moreover, grade 3 side effects were no longer observed at 20-week follow-up (Table 4). In patients from the imiquimod group who completed the assigned treatment, the protocol of applying imiquimod at least twice per week for 16 weeks was strictly followed by 81.4% (35/43) of women, and the treatment was successful in 60.0% (21/35) of cases. Others either applied it once per week for a certain period of time, periodically discontinued treatment, or applied it irregularly. In all cases where patients decided to lower the frequency of applying imiquimod, this was due to its side effects. There were only 18.6% (8/43) of patients who used imiquimod irregularly or had temporarily stopped the treatment. However, even in this group, the success rate was substantially high, since treatment was successful in 62.5% (5/8) of the patients. The menstrual cup as an applicator was acceptable for 84.6% (44/52) of the patients; 11.5% (6/52) of the patients had difficulties in removing the applicator; 1.9% (1/52) found it unpleasant, and only 1.9% (1/52) thought that the method of application was unacceptable or inappropriate. All nine patients in the imiquimod group who discontinued treatment completely and did not complete the sixteen-week protocol did so in the first 10 weeks of the study, six of them due to side effects, one due to suspected invasion; one did not show up for the scheduled follow-ups, and one thought that this method of treatment was unacceptable. We performed LLETZ in all these patients, and confirmed HSIL in final histology in 87.5% of the cases. Between the 10th and 16th week of imiquimod application, we had no cases of complete discontinuation of the treatment. Owing to the high incidence of side effects, there was a substantially higher workload with unplanned visits and email and telephone consultations. For example, there were 509 additional corresponding emails between doctors and patients.

In the LLETZ group, treatment was completed by 100% (52/52) and was successful in 92.3% (48/52) of the patients. In the CIN2p16+ and CIN3 subgroups, treatment was successful in 84.2% (16/19) and 97.0% (32/33) of patients, respectively (Table 2). Side effects were present in 44.2% (23/52) of patients (Table 3, Appendix A). Moderate (grade 2) side effects were present in 13.5% (7/52) of patients. We did not observe any severe (grade 3) side effects (Table 3). Total excision with free margins was achieved in 73.0% (38/52); positive margins were present in 13.5% (7/52), and in 13.5% (7/52) of the cases, fragmentation or extensive thermal artefacts interfered with pathological interpretation. When comparing these three groups (negative margins vs. positive margins vs. nondefined margins), there was no difference in treatment success (92.1% vs. 100% vs. 85.7%, *p* = 0.385).

Treatment with LLETZ was significantly more successful than treatment with imiquimod (92.3% vs. 51.9%; *p*-value < 0.001) and was associated with fewer side effects overall, as well as both moderate and severe side effects (44.2% vs. 88.5%, 13.5% vs. 38.5% and 0% vs. 13.5%, respectively; *p*-value < 0.001). However, in the subgroup of CIN2p16+ patients, there were no statistically significant differences between groups (84.2% vs. 73.9%; *p*-value = 0.477) regarding treatment success, although the absolute difference in success rates was more than 10% (Table 2). No cases of progression to cancer and no grade 4 side effects were observed in either of these groups.

## 4. Discussion

In this prospective randomized trial, patients who underwent treatment with imiquimod had overall worse treatment success than those who underwent treatment with LLETZ (51.9% vs. 92.3%; *p*-value < 0.001). No progression to invasive disease was observed in either of these groups. Similar results regarding the overall success rate of treatment with imiquimod (59–83.3%) have been obtained in other published studies, our result (53%) being at the lower limit of results published so far [16,18,31,32]. This observation can be explained by predominantly high rates of moderate and severe side effects (38.5% and 13.5%) with 17.3% drop out of patients.

A post hoc subgroup analysis showed that in patients with CIN3, LLETZ was by no doubt associated with the better outcome. However, in the CIN2p16+ group, treatment with imiquimod was not inferior to treatment with LLETZ (73.9% vs. 84.2%; *p*-value = 0.424). Nevertheless, this finding should be interpreted with caution, and could be purely accidental, since (i) it is based on a post hoc subgroup analysis; (ii) the sample size calculation was based on HSIL, and not for the subpopulation of patients with CIN2p16+ lesions, and (iii) the number of patients in this subgroup was low.

The decision on how to define success of treatment was not easy because the published trials’ take on this issue varies greatly. Other study protocols often describe different outcome measures in both arms and define success of treatment as cytological or histological regression to CIN1 or less in the imiquimod group [16,18,19,31], as normal cytology either in LLETZ or in both groups [19,20], or as no need for second LLETZ after a six-month follow-up [18]. We believe that when comparing the two methods, either to prove interchangeability of the methods or to compare a new treatment modality to the standard treatment, it is crucial to have the same outcome measure in both groups. With this in mind, we defined the success of treatment of HSIL as an absence of HSIL at the end of the treatment period in both groups, assessed by either biopsy or cytology, since the presence of low-grade lesions (CIN1, CIN2p16-) may be persistent without requiring further treatment. It is also important that patients were followed-up in an experienced colposcopy centre, because biopsies are crucial in determining the success of treatment with imiquimod, as inter- and intraobserver variability is common in colposcopy, cytology, and pathology, causing different predictive values of the methods [33]. The highest predictive values for diagnostics of HSIL therefore occur within colposcopy units with a team dedicated to cervical pathology and having regular prospective clinical audits [34,35]. Our results from previous clinical audits showed that we observed a high number of nondefined margins after LLETZ, as pathologists often describe the margins as too damaged to be interpreted. We also observed higher-than-expected numbers in the prevalence of low-grade, abnormal cytopathological results (ASC-US or LSIL) after LLETZ. However, both of these cyto- and pathological characteristics are rarely associated with the need for reoperation, which is only necessary in 2.9% of the cases and comparable to that of other centres [36].

The advantage of treatment with imiquimod is that it can be performed at home, using a vaginal applicator or, in our case, a menstrual cup, which was well-tolerated by 84.6% of patients. Imiquimod was used at least twice per week in 81.4% of patients. There was no difference in the success of the treatment between groups that used imiquimod regularly or irregularly. Considering these results and the fact that almost 20% of women could not follow the designed application protocol, we propose that imiquimod application twice per week might also be sufficient to achieve high treatment success rates by reducing the prevalence and duration of adverse side effects. However, further randomized controlled trials will be necessary to confirm this thesis. Beside a lower overall success rate, another disadvantage of treatment with imiquimod is the occurrence of side effects. Side effects and severe side effects (grade 2 and 3) were significantly more common in the imiquimod group compared to the LLETZ group (88.5% vs. 44.2% and 51.9% vs. 13.5%, respectively) and were the main reason for complete treatment discontinuation in the experimental arm. Overall, we know from previous data that treatment with imiquimod can be associated with a great number and variety of predominantly short-term side effects [18,37,38], whereas LLETZ is associated with short- and long-term side effects [10,11,12,13,14,39,40]. In the group of patients treated with imiquimod that were able to complete the treatment, we observed that the severity of side effects did not increase during the course of treatment. Quite the opposite, after 10 weeks, there was a decrease in grade 1 and 2 side effects and a complete disappearance of grade 3 side effects. However, given the high incidence of side effects in the imiquimod group, there were substantially more unscheduled outpatient visits and telephone and email consultations, which also need to be taken into account. Moreover, we believe that especially severe side effects demotivated the patients to comply with the treatment, despite fairly good treatment results. It is also important to emphasize that the majority (87.5%) of patients who completely discontinued treatment at 10 weeks had HSIL in histology after LLETZ. Therefore, we would not advise the duration of treatment with imiquimod to be shorter than 16 weeks. In the future, the main challenge will be to find a treatment protocol that could combine the highest possible efficacy, tolerability, and side effect profile rate with the lowest possible side effect rate.

Regarding other conservative, nonsurgical treatments, it would be interesting to compare the treatment of cervical lesions with imiquimod to photodynamic therapy, cold-coagulation, CO_2_ laser, or others. The effects of cold-coagulation [41] and CO_2_ laser [42] have already been well-established and there are also studies published about the moderate or fairly good effect of photodynamic therapy on cervical lesions [43,44], and the effect of combined therapy for verrucous carcinoma and vulvar intraepithelial neoplasia [45,46]. In the future, there is, therefore, an option to either combine treatments for cervical lesions or to compare them to one another.

There are several advantages of our study. Firstly, it was a randomized controlled trial. All patients were treated in an outpatient setting, and close follow-up at a specialized colposcopy clinic was provided for all of them. However, when interpreting the results of our study, it should be kept in mind that the participants represent a selected group of newly diagnosed, previously untreated, younger patients with colposcopy. The data on additional secondary outcomes of our trial, such as the need for reoperation in two years and the modulatory effect of imiquimod on immunoregulatory molecules, are not yet available and are expected in three years’ time. The data concerning the need for reoperation in two years is very important, as it will provide information on safety of imiquimod as well as comparable intermediate-time efficacy compared to that of LLETZ as the gold standard.

There are also some limitations of our work. Firstly, the study was only able to provide short-term results, as the follow-up period from the start of the treatment was 20 weeks in the experimental arm and six months in the control arm. Additionally, women in the imiquimod group had shorter follow-up, which may affect the outcome. However, we aim to overcome this limitation by following these patients for a longer period. Secondly, the sample-size calculation was performed for patients with HSIL, meaning that we have to interpret our findings on patients with CIN2p16+ lesions with extreme caution. Moreover, almost 20% of patients failed to administer imiquimod as prescribed or did not complete the assigned treatment. In order to avoid this bias, ITT analysis was performed when comparing the data on treatment success. Furthermore, women in the imiquimod group underwent colposcopy and had biopsies collected, whereas women who had an excision only had a smear. This may have increased the risk of verification bias.

## 5. Conclusions

Our results confirmed that LLETZ remains the gold standard for treating HSIL. Treatment with imiquimod offered a moderate success rate and was associated with a clinically important rate of side effects and severe side effects, causing patients to discontinue treatment. According to our results, we would strongly suggest conducting a new trial in the subgroup of patients with intermediate-risk HSIL (CIN2p16+), where we showed non-inferiority of imiquimod treatment to LLETZ; however, the power of our results is not sufficient to make firm conclusions. Conservative treatment would be beneficial in this group of patients, especially when it comes to younger patients. Furthermore, larger studies evaluating the long-term effects of this treatment are needed, especially in view of disease progression, recurrence, and HPV clearance.

## Figures and Tables

**Figure 1 jcm-10-05777-f001:**
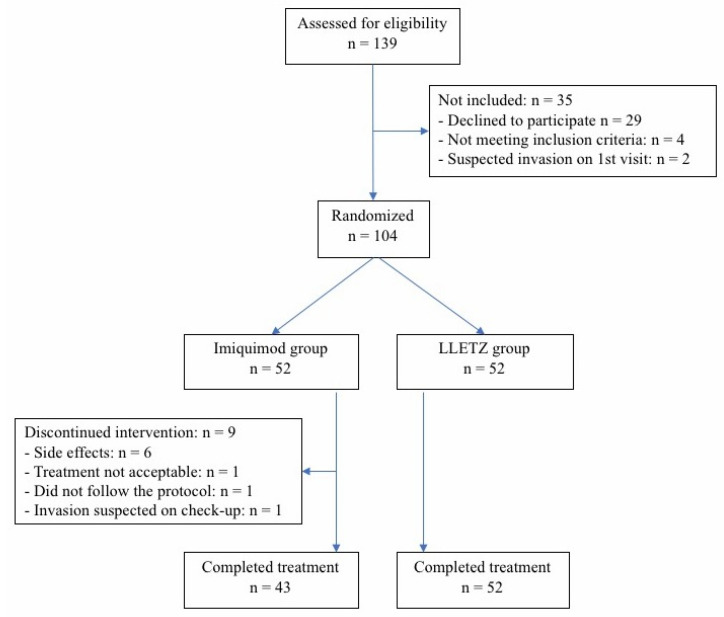
Flow diagram of the study.

**Table 1 jcm-10-05777-t001:** Basic patients’ characteristics.

Variable		All Patients *n* = 104	Group 1 (Imiquimod) *n* = 52	Group 2 (LLETZ) *n* = 52	*p*-Value
Age (mean ± SD) Smoking: yes (*n* (%))		28 ± 4.4	28.3 ± 4.2	27 ± 4.6	0.52
		39 (37.5)	22 (42.3)	17 (32.7)	0.42
Histopathological diagnosis	CIN2p16+ (*n* (%))	42 (40.4)	23 (44.2)	19 (36.5)	0.55
	CIN3 (*n* (%))	62 (59.6)	29 (55.8)	33 (63.5)	

**Table 2 jcm-10-05777-t002:** Success of treatment in patients treated with imiquimod (Group 1) and LLETZ (Group 2).

	Group 1 (Imiquimod)	Group 2 (LLETZ)	*p*-Value
All patients (*n* /total *n* (%))	27/52 (51.9)	48/52 (92.3)	<0.001 *
CIN 2 (*n* /total *n* (%))	17/23 (73.9)	16/19 (84.2)	0.477
CIN 3 (*n* /total *n* (%))	10/29 (34.5)	32/33 (97.0)	<0.001 *

* Statistically significant difference. LLETZ, large-loop excision of the transformation zone.

**Table 3 jcm-10-05777-t003:** Presence of side effects and the highest grade of side effect (using the 5th version of CTCAE guidelines) in patients treated with imiquimod (Group 1) and LLETZ (Group 2).

		Group 1 (Imiquimod) *n* = 52	Group 2 (LLETZ) *n* = 52	*p*-Value
Side effects: yes (*n* (%))		46 (88.5)	23 (44.2)	<0.001 *
Highest grade of side effect (*n* (%))	Grade 1	18 (34.6)	14 (26.9)	<0.001 *
	Grade 2	20 (38.5)	7 (13.5)	
	Grade 3	7 (13.5)	0 (0)	
	Other	1 (1.9)	2 (3.8)	

* Statistically significant difference. LLETZ, large-loop excision of the transformation zone.

**Table 4 jcm-10-05777-t004:** Side effects in the experimental (imiquimod) arm.

	Before 10 Weeks (*n* = 52)	After 10 Weeks (*n* = 43)	Any Time (*n* = 52)
Side effects: yes (*n* (%))	42 (80.8)	33 (76.7)	46 (88.5)
Grade 1 (*n* (%))	35 (67.3)	26 (60.5)	36 (69.2)
Grade 2 (*n* (%))	18 (34.6)	14 (32.6)	21 (40.4)
Grade 3 (*n* (%))	7 (13.5)	0 (0)	7 (13.5)
Grade 4 (*n* (%))	0 (0)	0 (0)	0 (0)
Other (*n* (%))	18 (34.6)	14 (32.6)	25 (48.8)

## Data Availability

Our data is confidential and hence not available.

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
