# Peer review of "Comparison of Conservative Treatment of Cervical Intraepithelial Lesions with Imiquimod with Standard Excisional Technique Using LLETZ: A Randomized Controlled Trial"

_jcm, 2021, doi:10.3390/jcm10245777_

Round 1

Reviewer 1 Report

Accept in present form

Reviewer 2 Report

As said in my last review I think this is a report on a very intersting prospective study. 

This manuscript is a resubmission of an earlier submission. The following is a list of the peer review reports and author responses from that submission.

Round 1

Reviewer 1 Report

It is a pity that during the diagnosis of the abnormal cytology result and after the examination period - 20 weeks, the HPV genotyping test was not performed. It is possible that statistical significance would be found between types 16, 18, 31 and others with less oncogenic potential.

In the case of the experimental study, I believe that the histopathological examination of the material from the endocervical canal should be obligatory both at the time of qualification for the study and after the observation period - 20 weeks, even in the case of adequate colposcopy and transformation zone type 1.

We should use the terminology of colspocopy adequate, not satisfactory.

How large percentage of patients diagnosed with HSIL decide to vaccinate - Gardasil 9?

Author Response

Dear Reviewers, thank you both very much for your valuable comments. We have made corrections of the manuscript and we also provided answers to your specific questions.

Reviewer A:

Comment A1:

It is a pity that during the diagnosis of the abnormal cytology result and after the examination period - 20 weeks, the HPV genotyping test was not performed. It is possible that statistical significance would be found between types 16, 18, 31 and others with less oncogenic potential.

Answer A1:

We agree with your observation. When we designed the study, we wanted the protocol in the imiquimod arm to be as close as possible to the national guidelines for standard treatment with LLETZ. Based on our national guidelines (mentioned in the line 100), we do not test HSIL patients for HPV but the follow up with HPV testing is performed in the next 2 years after the procedure (once yearly). For that reason, we did not include the HPV testing in our present study. However, in the ongoing long-term follow-up,  HPV testing is being performed and we will be able to stratify and differentiate our patients based on different HPV genotypes.  

Comment A2:

In the case of the experimental study, I believe that the histopathological examination of the material from the endocervical canal should be obligatory both at the time of qualification for the study and after the observation period - 20 weeks, even in the case of adequate colposcopy and transformation zone type 1.

Answer A2:

Since we do not routinely perform ECC in cases with adequate colposcopy, we did not include this in our study. However, we agree that it should be included and we will consider this in our future studies. In the present case, we unfortunately can not address this.

Comment A3:

We should use the terminology of colposcopy adequate, not satisfactory.

Answer A3:

Corrected in lines 104 and 325.

Comment A4:

How large percentage of patients diagnosed with HSIL decide to vaccinate - Gardasil 9?

Answer A4:

In our country, HPV vaccination of girls started in the year 2009. Girls are vaccinated in the 6th grade of elementary school and the vaccination is free of charge. None of the patients included in the study were previously vaccinated, since they were all older. We are currently offering all our patients vaccination after treatment, especially if they are HPV negative. We are currently still gathering the data for our cohort, which will be published after the end of the follow-up (in about 1 year time).

Reviewer 2 Report

The manuscript “Comparison of Conservative Treatment of Cervical Intraepithelial Lesions with Imiquimod with Standard Excisional Technique Using LLETZ: A Randomized Controlled Trial.” reports very interesting data.

In this prospective trial patients with HSIL were randomized to either treatment with LLETZ (control arm) or imiquimod (IMQ, experimental arm). Although the data are interesting, the manuscript is poorly written, and in some cases difficult to understand. I suggest to submit the manuscript to proof reading, and in a next step to resubmit the paper.  

Please find enclosed some suggestions to improve the manuscript.

Suggestions:

Abstract:

Sentence in lines 30-31: Conclusion: In this trial, we showed that, in a general population of HSIL, LLETZ

remains the gold standard for treatment. Should be. ... Generally, in patients with HSIL, LLETZ remains the gold standard of treatment.

Introduction:

Lines 39-40, should be the incidence of and the mortality from...

Please check references. In line 66 you state that IMQ has promising effects on cervical cancer – however, the reference you are referring to is a paper about vulvar intraepithelial neoplasia

Line 72 please check whether local is the correct word

Results:

There seems to be a formatting issue with table 1

Discussion:

As adverse events and severe adverse events are the most important reason for treatment discontinuation, I would suggest to include a list of adverse events that have happened into the main results.

Performing a proper sample size calculation during statistical planning is not a advantage of a study but rather a necessity.

Since CIN2p16+ is a part of HSIL, your analysis regarding these patients is just a post-hoc subgroup analysis.

Author Response

Dear Reviewers, thank you both very much for your valuable comments. We have made corrections of the manuscript and we also provided answers to your specific questions.

Reviewer B

We did perform a proof reading of the text by an English native speaker, however we did make some changes afterwards, which could affect style and language. We tried to do additional proof reading and actually got the confirmation from the same professor, but due to holidays she unfortunately could not revise the text until 5th of November (due date for the revision). We will ask the journal for the final proof reading and we made changes in the text according to your suggestions.

Comment B1:

Abstract:

Sentence in lines 30-31: Conclusion: In this trial, we showed that, in a general population of HSIL, LLETZ

remains the gold standard for treatment. Should be. ... Generally, in patients with HSIL, LLETZ remains the gold standard of treatment.

Answer B1:

Corrected.

Comment B2:

Lines 39-40, should be the incidence of and the mortality from…

Answer B2:

Corrected.

Comment B3:

Please check references. In line 66 you state that IMQ has promising effects on cervical cancer – however, the reference you are referring to is a paper about vulvar intraepithelial neoplasia.

Answer B3:

We apologize for this mistake. It is corrected.

Comment B4:

Line 72 please check whether local is the correct word.

Answer B4:

We deleted the word local as we agree it is unnecessary.

Comment B5:

There seems to be a formatting issue with table 1

Answer B5:

Corrected. We formatted the table.

Comment B6:

As adverse events and severe adverse events are the most important reason for treatment discontinuation, I would suggest to include a list of adverse events that have happened into the main results.

Answer B6:

Thank you for the observation and suggestion. We also thought of this and actually made a version of the manuscript containing all the adverse effects, but since there were so many different ones observed (around 60), we decided to put all the adverse effects in the table in the supplement. If you think it would be better to put all of them in the text, we will modify the manuscript.

Comment B7:

Performing a proper sample size calculation during statistical planning is not a advantage of a study but rather a necessity.

Answer B7:

We deleted this statement.

Comment B8:

Since CIN2p16+ is a part of HSIL, your analysis regarding these patients is just a post-hoc subgroup analysis.

Answer B8:

Corrected and clarified in the lines 174 and 250.